# 10-year multimorbidity patterns among people with and without rheumatic and musculoskeletal diseases: an observational cohort study using linked electronic health records from Wales, UK

Farideh Jalali-najafabadi [ID],[1] Rowena Bailey,[2] Jane Lyons [ID],[2] Ashley Akbari [ID],[2] Thamer Ba Dhafari,[3] Narges Azadbakht,[3] James Rafferty,[2] Alan Watkins [ID],[2] Glen Philip Martin [ID],[3] John Bowes,[1,4] Ronan A. Lyons [ID],[2] Anne Barton [ID],[1,4] Niels Peek[3,5]

For numbered affiliations see end of article.

**Correspondence to**
Dr Farideh Jalali-najafabadi;
farideh.jalali@manchester.ac.uk

## ABSTRACT

**Objectives** To compare the patterns of multimorbidity between people with and without rheumatic and musculoskeletal diseases (RMDs) and to describe how these patterns change by age and sex over time, between 2010 and 2019.

**Participants** 103 426 people with RMDs and 2.9 million comparators registered in 395 Wales general practices (GPs). Each patient with an RMD aged 0–100 years between January 2010 and December 2019 registered in Clinical Practice Research Welsh practices was matched with up to five comparators without an RMD, based on age, gender and GP code.

**Primary outcome measures** The prevalence of 29 Elixhauser-defined comorbidities in people with RMDs and comparators categorised by age, gender and GP practices. Conditional logistic regression models were fitted to calculate differences (OR, 95% CI) in associations with comorbidities between cohorts.

**Results** The most prevalent comorbidities were cardiovascular risk factors, hypertension and diabetes. Having an RMD diagnosis was associated with a significantly higher odds for many conditions including deficiency anaemia (OR 1.39, 95% CI (1.32 to 1.46)), hypothyroidism (OR 1.34, 95% CI (1.19 to 1.50)), pulmonary circulation disorders (OR 1.39, 95% CI 1.12 to 1.73) diabetes (OR 1.17, 95% CI (1.11 to 1.23)) and fluid and electrolyte disorders (OR 1.27, 95% CI (1.17 to 1.38)). RMDs have a higher proportion of multimorbidity (two or more conditions in addition to the RMD) compared with non-RMD group (81% and 73%, respectively in 2019) and the mean number of comorbidities was higher in women from the age of 25 and 50 in men than in non-RMDs group.

**Conclusion** People with RMDs are approximately 1.5 times as likely to have multimorbidity as the general population and provide a high-risk group for targeted intervention studies. The individuals with RMDs experience a greater load of coexisting health conditions, which tend to manifest at earlier ages. This phenomenon is particularly pronounced among women.

## STRENGTHS AND LIMITATION OF THIS STUDY

⇒ Our large study cohort provides sufficient statistical power to calculate overall and age-specific and sex-specific prevalence rates of individual comorbidities between 2010 and 2019.

⇒ In addition to the recognised and frequently reported comorbidities such as cardiovascular risk factors, our findings highlight several comorbidities such as deficiency anaemia, fluid and electrolyte disorders and valvular disease that are highly prevalent in people with RMDs, but which are under-reported in the literature. We prioritise the early identification of RMDs and related health conditions, with a particular focus on younger women.

⇒ We have analysed only pairs of diseases, but higher-order combinations of diseases probably interact in more complicated ways. In terms of the application of these findings, it should be noted that the prevalence measure was based on Elixhauser conditions; there is no established gold standard list of diagnoses used to define multimorbidity and definitions of multimorbidity vary in the medical literature.

⇒ The origins of RMD in Wales are probably similar to those in other UK nations with comparable access to the National Health Service and genetic links related to RMD. As a result, we expect that the risk factors and the prevalence of coexisting health conditions will exhibit similarities. However, when applying this approach to other countries, it is crucial to recognise that the specific comorbidities may vary based on the distinct health characteristics of each nation. To enhance the applicability of our findings, validation across diverse healthcare systems and patient populations is essential.

⇒ To our knowledge, this is the first study to present analysis using different approaches in a large population of Wales in people with and without rheumatic and musculoskeletal diseases (RMDs) using routine primary care data.

Additionally, there is an under-reporting of comorbidities in individuals with RMDs.

## INTRODUCTION

Multimorbidity is a global public health concern and refers to the existence of multiple long-term conditions in a single individual.[1–3] Morbidity refers to individual long-term morbid conditions. The prevalence of multimorbidity appears to have increased in many regions of the world over the past 20 years. It is anticipated to continue to rise and evidence from high-income countries suggests that, while multimorbidity is highly prevalent in older populations (typically those over 65 years of age), it also affects younger people.[4] Epidemiologically, the term comorbidity is used when estimating co-occurring conditions among populations with an index condition and the term 'multimorbidity' is used when estimating the prevalence of co-occurring conditions among the wider population.[5 6]

Rheumatic and musculoskeletal diseases (RMDs) comprise over 200 diseases and syndromes, which can affect the joints, connective tissue, cartilage and tendons; they are usually progressive, associated with pain and are among the most common chronic non-communicable diseases. The prevalence of disability, premature mortality and physical impairment[7 8] in RMDs is increased meaning they are a major burden on individuals, social care systems and health systems and this burden has been recognised by WHO and the United Nations.[9] For example, in the UK, an estimated 8.75 million people have sought treatment for osteoarthritis, the equivalent to a third of all people over 45 years of age.[10] In addition, an estimated one in five men and one in two women aged over 50 years will sustain a low-trauma fracture as a result of osteoporosis[11–13] with the situation set to worsen with ageing populations. Previous studies have found that people with RMDs are often living with additional chronic conditions[14–16] and RMDs appear to form a principal component of certain multimorbidity clusters.[17–19] Certainly, a substantial proportion of people with RMDs now live with multimorbidity,[20] but most current clinical guidelines have 'single-condition' model of care without considering multimorbidity. Multimorbidity has a number of negative outcomes, including disability and death, reduced quality of life and higher healthcare utilisation and expenses.[21]

Several studies have described comorbidities in the RMDs population,[22 23] but it remains to be determined if the increased comorbidity in patients with RMDs relates to specific comorbid conditions, occurs across the life course or differs by sex. The longitudinal nature of the (Secure Anonymised Information Linkage (SAIL)) Databank in Wales, UK, enables the analysis of information on diagnoses and consultations at multiple time points for RMDs in general.[24 25] Therefore, using a large electronic linked primary care data set on the resident population in Wales, this study aimed to examine the patterns of 28 chronic conditions in people with and without RMDs over a 10-year study period. We further stratified the comorbidities prevalence rates by age, gender and assessed the association between diagnosis of RMDs and the presence of a comorbidity and described the patterns of all two coexisting comorbidities.

## METHODS

### Data sources

Data used for this study were accessed via the SAIL Databank (www.saildatabank.com). SAIL is a state-of-the-art, remotely accessible, privacy-protecting Trusted Research Environment accredited under the Digital Economy Act. SAIL contains linked anonymised population scale data sources at an individual person level, household level and multiple ecological levels, for the population of Wales.[24 25] SAIL Databank sources include inpatient, outpatient, critical care, emergency department, mortality, birth register and demographic information for people that use healthcare services in Wales.

Morbidities were defined based on Elixhauser categories, a well-established scheme distinguishing 30 classes of chronic diseases. HIV was not included as data is not available in SAIL due to data privacy policies. The primary care Read codes V.2 (CTV2) were used to identify Elixhauser morbidities in healthcare records.

The Welsh Multimorbidity e-Cohort (WMC) is based on the 83% of the Welsh population covered by general practice (GP) practices in Wales that contribute to the SAIL Databank. A report by Jones in 2022[26] (online supplemental material) showed how this compared with the 100% of the population. The distributions of sex and age group are identical, and the distribution of deprivation scores are very similar. The Elixhauser coding algorithms used to establish the presence of each Elixhauser morbidity were downloaded from online supplemental material of Metcalfe et al[27] and resulted in a set of 28 comorbidities and rheumatoid arthritis/collagen vascular diseases (RMDs).

### Patient and public involvement

None

### Missing data

This analysis used age, sex and the presence or absence of comorbidities. Age and sex were complete as they were derived from the algorithm creating the anonymised linkage fields from National Health Service (NHS) registration data. The presence of comorbidities was defined by the presence of the relevant diagnostic codes in the electronic health records. When there was no diagnostic code for a particular condition in a patient's record, it was assumed that that patient did not have that condition. This is a common approach when using data from electronic health records. As a result, there were no missing data in the data that we analysed.

### Participants

Study participants were selected from the WMC, which consists of 2 902 101 million people living in Wales on 1

January 2000 with follow-up until 31 December 2019 and was created to provide accessible research ready data for the understanding of multimorbidity.[28]

## RMDs cohort

Patients aged 0–100 years with a first diagnostic RMDs medical code between 1 January 2010 and 31 December 2019 were identified. The index date was defined as the earliest date of RMDs diagnosis. Patients were followed until movement out of a SAIL providing practice, death or end of the study observation period (31 December 2019). Our sample was restricted to patients having≥1-year follow-up for each matched case and comparator (eg, cases and comparators in the matched cohort in 2019).

## Non-RMD cohort

We performed a matched, retrospective cohort study within the WMC. Within each year, up to five patients without RMDs (comparators) were identified and matched to each patient with RMDs on the following variables at the RMDs diagnosis date: age (±2 years difference), sex and GP code.

A time period of ±2 year was used for matching to increase the comparator sample after discussion with clinicians that age groups within this interval are clinically comparable. CTV2 for GP records were used to identify patient records with an RMD diagnosis. We required each comparator to be enrolled with the same GP on the index date of the accompanying patient with RMDs and assigned the same index date.

## Outcomes

Overall, the presence of morbidities was defined as conditions recorded before and after RMDs diagnosis date (index date) and up to the end of the calendar year. We defined multimorbidity as the presence of at least two conditions from the Elixhauser comorbidity index, excluding RMDs as one of the conditions, to avoid bias in our results towards greater multimorbidity in the RMDs cohort.

## Statistical analyses

Descriptive statistics of the differences between the cohorts are reported as the total percentage of multi morbid persons (two or more) conditions, the percentage distribution of the total number of comorbidities and the mean number of comorbidities by sex and age group. To explore the patterns of multimorbidity, the data were analysed using four approaches:

► Temporal trends for specific Elixhauser conditions were explored using prevalence rates. Annual prevalence rates were derived for each cohort over the 10-year period and were calculated separately for each comorbidity by dividing the number of persons diagnosed in or before the study year by the total number of persons contributing throughout the same study year. Corresponding 95% CIs were calculated.

► The prevalence rates were also calculated stratified by: (1) 20-year age groups and (2) sex, with RMDs and matched comparator patients in the last study year.

► The patterns of morbidity conditions were examined using pairwise associations between all comorbidities. The percentage of patients with both comorbidities was calculated for each pair of conditions for each cohort.

► Logistic regression models were used to explore the association between the presence/absence of each comorbidity and diagnosis of RMD. Conditional models, adjusted for age and sex, were fitted to estimate differences between cohorts with associations for Elixhauser comorbidities. The coefficients were exponentiated and reported as ORs along with corresponding 95% CIs. Regression models restricted to males and females were also fitted to calculate sex-specific estimates.

All analyses were conducted using Python software V.3.7. We used the Strengthening the Reporting of Observational Studies in Epidemiology cohort checklist when writing our report.[29]

## RESULTS

### Patient population

Of 2 902 101 million individuals enrolled in WMC between 2000 and 2019, we identified 102 042 who fulfilled our eligibility criteria for people with RMDs. Patients with RMDs were matched to 1 637 632 million people without RMDs (comparators). Figure 1 shows the flow chart for identification of patients with and without RMDs registered in linked GPs.

### Annual prevalence of comorbidities

We investigated the prevalence of 28 comorbidities overall, but results for 7 comorbidities with less than 5 subjects have not been reported due to statistical disclosure control requirements in SAIL Databank. Statistical disclosure control rules meant that cells with less than five subjects cannot be reported in order to prevent the inadvertent disclosure of individuals. The UK's Office for National Statistics (ONS) created guidance (https://ukdataservice.ac.uk/app/uploads/thf_datareport_aw_web.pdf) to safeguard privacy through design and statistical disclosure control. SAIL adheres to the ONS disclosure regulations.

The annual prevalence rates of the remaining 21 comorbidities between 2010 and 2019 are presented in figure 2. The most common comorbidities were cardiovascular risk factors, hypertension and diabetes without complications. Other neurological disorders (OND), increased sharply over the study period in people with RMD from 0.9% (95% CI: 0.56% to 1.24%) in 2010 to 10.2% (95% CI: 9.86% to 10.62%) in 2019; and from 0.7% (95% CI: 0.56% to 0.83%) in 2010 to 8.6% (95% CI: 8.41% to 8.73%) in 2018 in comparators. Drug abuse, peptic ulcer disease and liver diseases were the least common chronic

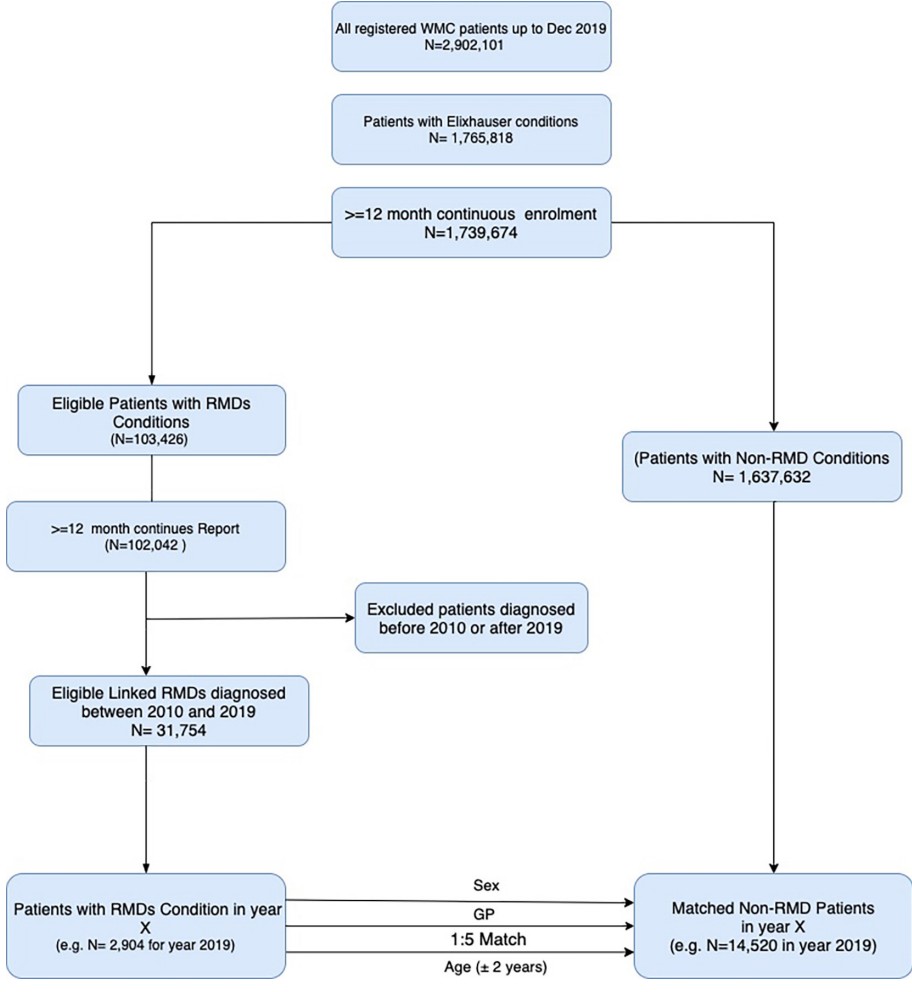

**Figure 1** Flow chart for identification of patients with and without RMDs registered in WMC, using year 2019 as an example. RMDs, rheumatic and musculoskeletal diseases; WMC, Welsh Multimorbidity e-Cohort.

conditions in both people with and without RMDs. The prevalence rates (and CIs) of all remaining 21 comorbidities in RMDs and comparators in 2010 and 2019 are presented in online supplemental table 1.

Rheumatoid arthritis (RA), connective tissue diseases, polymyalgia rheumatica and giant cell arteritis and juvenile idiopathic arthritis are different type of RMDs. For all these subtypes similar to the RMD group as a whole, we found the most common comorbidities to be cardiovascular risk factors, hypertension and diabetes without complications.

The annual prevalence of Elixhauser comorbidities for each RMDs subtype between 2010 and 2019 is presented in online supplemental figure 1.

### Number of comorbidities, age specific, sex specific
In 2019, patients with RMDs had a higher proportion of multimorbidity (two or more conditions in addition to the RMDs) compared with the comparator cohort (81% and 73%, respectively) figure 3A. The mean number of comorbidities increased with age for both groups but was higher in women with RMDs between the ages of 25 and 100 compared with the comparator group while in men, the mean number of comorbidities was higher

among patients with RMD only from the age of 50 years, figure 3B.

The age–gender-specific analysis, in relation to the specific comorbidities for RMDs and comparators, are presented in 2019, figure 4. Depression was observed as the most prevalent condition in the age groups 16–50 years for RMDs and non-RMDs, with diabetes without complications being the second most prevalent condition in patients with RMDs in those age groups. OND and cancer were the second and third most prevalent condition for patients with non-RMD, respectively. In 2019, the prevalence of hypertension was nearly 4% higher in women and men than comparators. The plot for other years in age–gender-specific analysis is presented in online supplemental figures 2–4.

### Proportion of patients with two comorbidities
The most common pairs of coexisting morbidities across both the RMD and comparator groups in 2019 were combinations of hypertension, chronic pulmonary disease, depression, diabetes and obesity. The patterns of all two coexisting comorbidities in people with and without RMDs in 2019 are illustrated in figure 5. The dark blue colour in the heatmap is

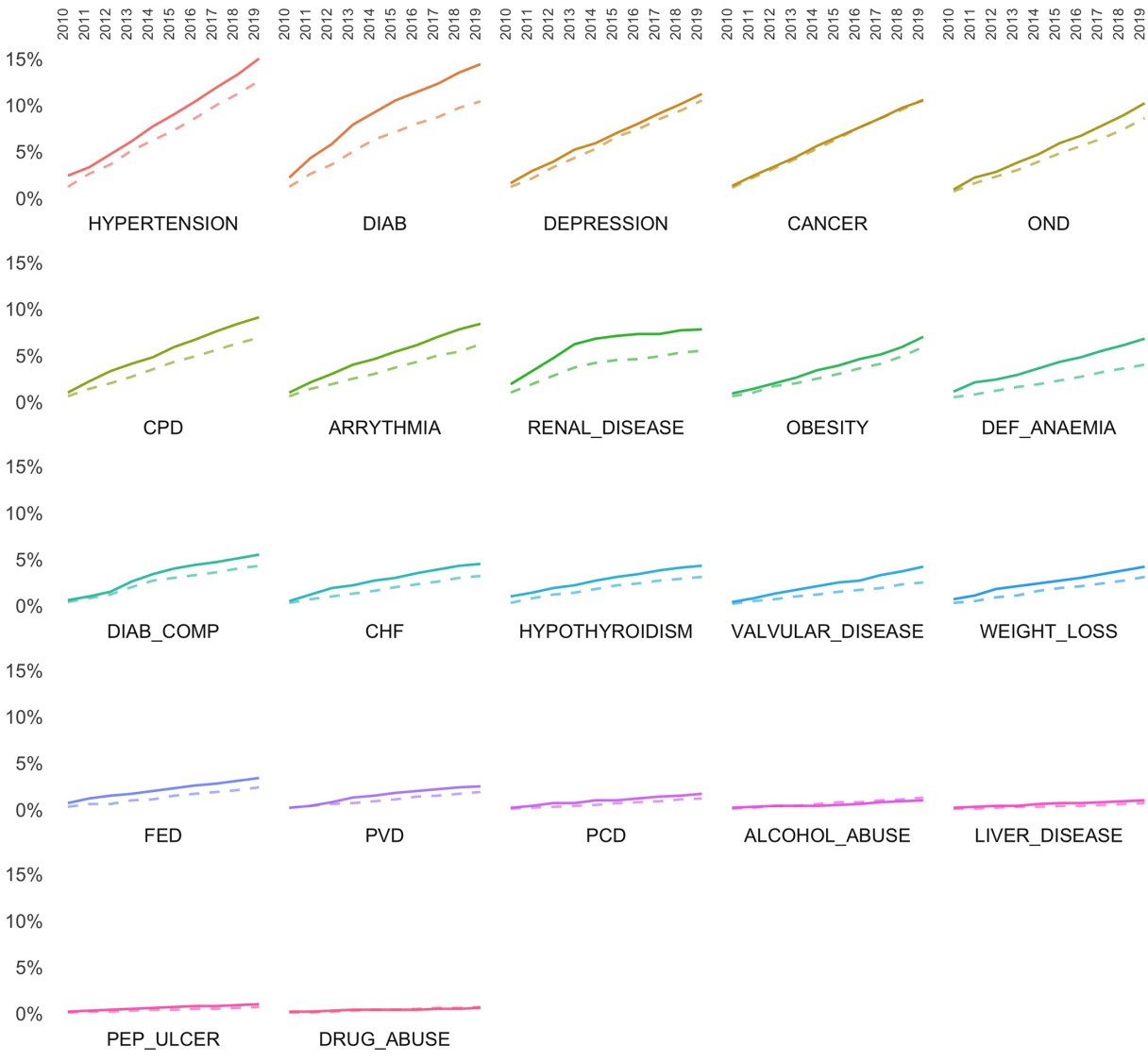

Prevalence of Elixhauser comorbidities in patients with RMDs and patients without RMDs between 2010 and 2019

— RMDs - - Non-RMDs

**Figure 2** Annual prevalence of Elixhauser comorbidities in RMDs cases and matched comparators 2010–2019. CHF, congestive heart failure; CPD, chronic pulmonary disease; DEF, deficiency ; DIAB, diabetes; DIAB_COMP, diabetes complicated; FED, fluid and electrolyte disorder; OND, other neurological disorders; PCD, pulmonary circulation disorders; PEP_ULCER, peptic ulcer disease excluding bleeding; PVD, peripheral vascular disorders; RMDs, rheumatic and musculoskeletal diseases.

determined by a higher percentage of patients with the two comorbidities in the same plot and not between separate plots. The plot for other years is presented in online supplemental figure 5.

### ORs (95% CI) for associations of RMDs with specific comorbidities in annual cohorts 2010–2019

Patients with RMDs were at least 30% more likely to have: deficiency anaemia (OR 1.39, 95% CI: 1.32 to 1.46), hypothyroidism (OR 1.34, 95% CI: 1.19 to 1.50), pulmonary circulation disorders (OR 1.39, 95% CI: 1.12 to 1.73), renal disease (OR 1.35, 95% CI: 1.22 to 1.50) and weight loss (OR 1.30, 95% CI: 1.17 to 1.45) compared with those without RMDs.

RMDs patients were also 27% more likely to have both and electrolyte disorders (OR 1.27, 95% CI: 1.17 to 1.38) and valvular disease (OR 1.27, 95% CI: 1.15 to 1.41), 17% more likely to have diabetes (OR 1.17, 95% CI: 1.11 to 1.23) and 9% more likely to have a diagnosis of obesity (OR 1.09, 95% CI: 1.05 to 1.13) when compared people without RMDs. A cancer diagnosis was 19% less likely for the RMDs group. Similarly, OND, diabetes with complications, peripheral vascular disorders, drug abuse and alcohol abuse were all significantly less likely to be diagnosed among people with RMDs compared with people without. ORs were generally stable over the study period between 2010 and 2019, with the exception

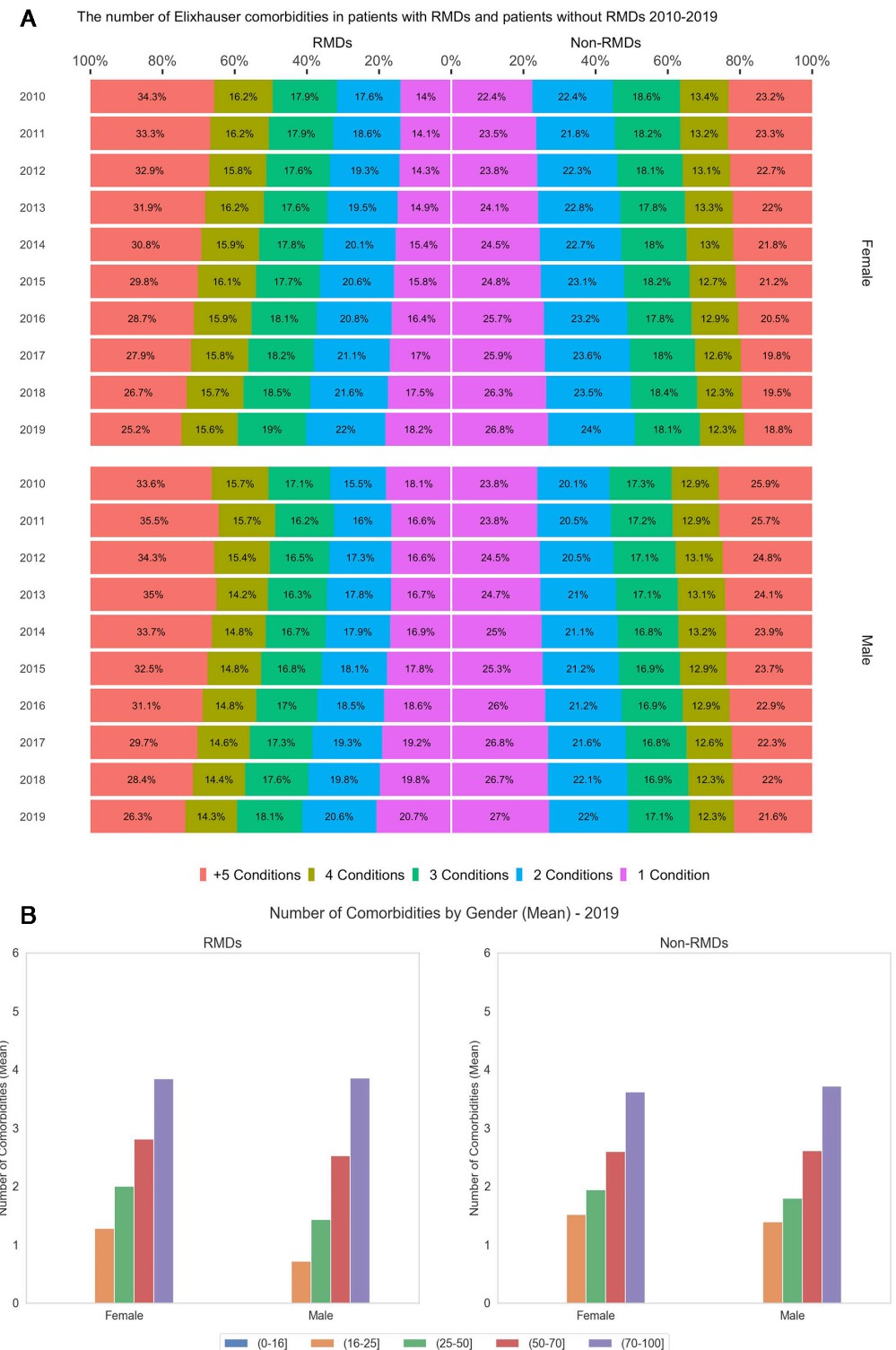

**Figure 3** Trends of comorbidities in patients with RMDs and matched comparators. (A) The number of conditions in patients with RMDs and matched comparators without RMDs in 2010–2019. (B) Mean number of comorbidities in patients with RMDs and comparators by age categories and sex 2019. RMDs, rheumatic and musculoskeletal diseases.

of depression, which was found to be significantly higher among patients with RMDs between 2010 and 2015 but was significantly (although slightly) lower in 2018 and 2019. The ORs are presented in table 1. The ORs for the gender-specific analyses is presented in online supplemental figure 6.

## DISCUSSION

To our knowledge, this is the first study to examine and compare the patterns of 28 Elixhauser-defined comorbidities using different statistical approaches in a large, well-characterised population of people with RMDs and people without RMDs in Wales. The primary care database

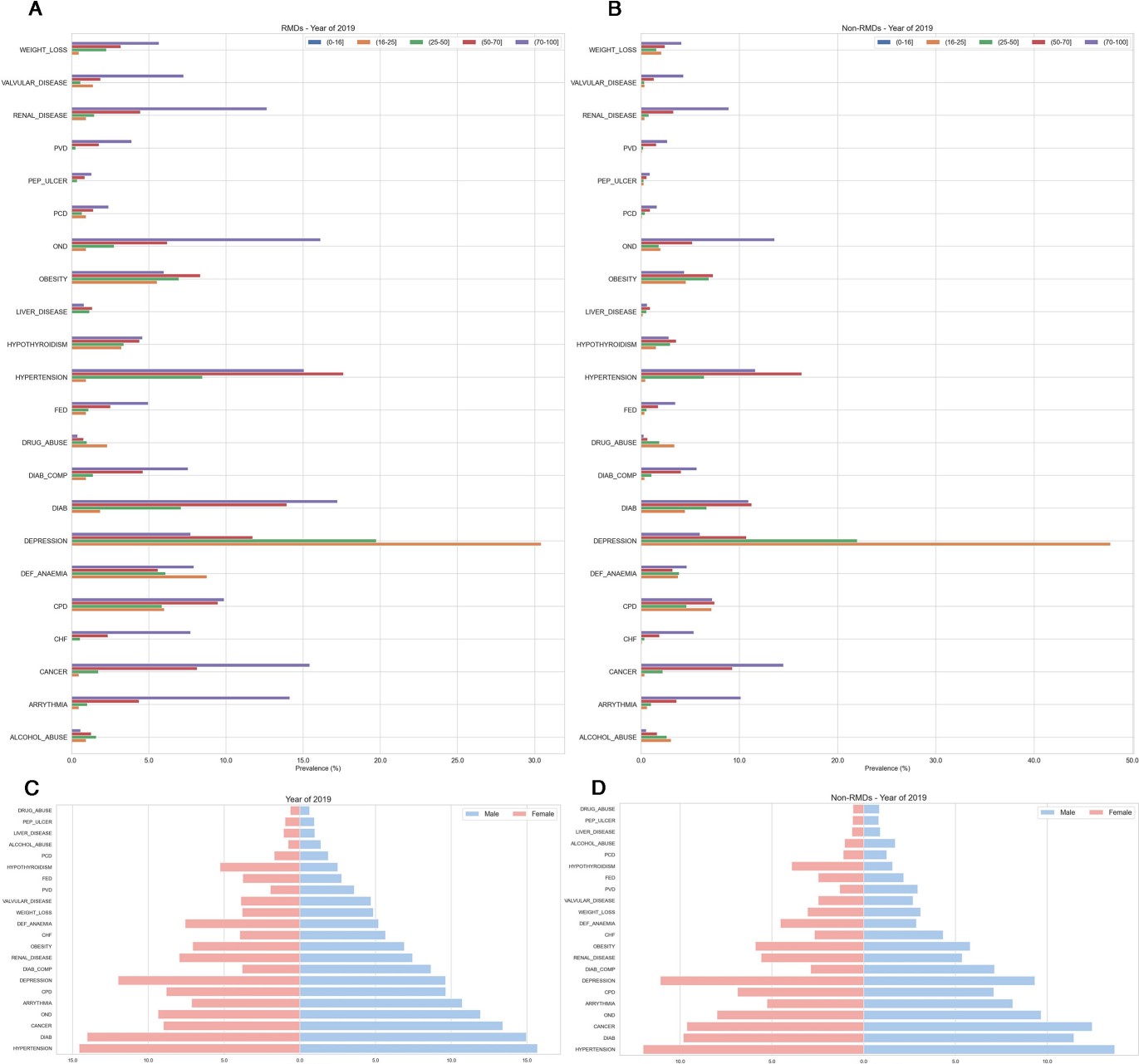

**Figure 4** Age–gender-specific analysis, in relation to specific Elixhauser comorbidities in 2019. (A) Age-specific analysis in patients with RMDs. (B) Age-specific analysis in comparators. (C) Gender-specific analysis in patients with RMDs. (D) Gender-specific analysis in comparators. CHF, congestive heart failure; CPD, chronic pulmonary disease; DEF, deficiency; DIAB, diabetes; DIAB_COMP, diabetes complicated; FED, fluid and electrolyte disorder; OND, other neurological disorders; PCD, pulmonary circulation disorders; PEP_ULCER, peptic ulcer disease excluding bleeding; PVD, peripheral vascular disorders; RMDs, rheumatic and musculoskeletal diseases.

that we used to estimate the prevalence of multimorbidity covered over 83% of the population in the Wales, so the findings should be generalisable in that population.

Our findings show that comorbidities were more prevalent in people with RMDs than comparators throughout the 10-year study period. Overall, comorbidities were more prevalent in women than in men and the prevalence rates increased with age. Our results demonstrate people with RMDs are approximately 1.5 as likely to have multimorbidity as the general population throughout

the 10-year study period. Diabetes and hypertension are the epicentre of RMD disease and potentially part of the trajectories of several other chronic conditions, suggesting a need for a more integrative multidisciplinary approach focusing on better management and prevention of these conditions.[2]

An Australian systematic review of population-based studies in the elderly[15] has previously reported that over 50% of elderly patients with arthritis also had hypertension, followed by cardiovascular diseases, dyslipidaemia,

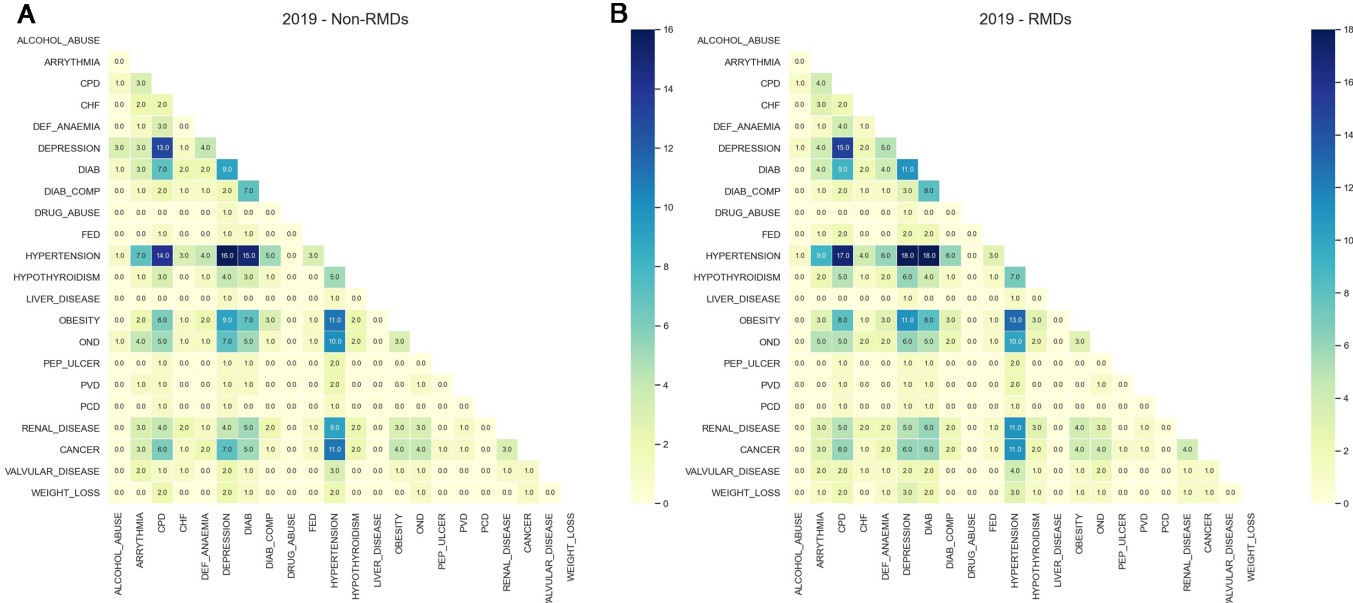

**Figure 5** Proportion of patients in relation to specific Elixhauser comorbidities in 2019. (A) RMDs. (B) Comparators. CHF, congestive heart failure; CPD, chronic pulmonary disease; DEF, deficiency; DIAB, diabetes; DIAB_COMP, diabetes complicated; FED, fluid and electrolyte disorder; OND, other neurological disorders; PCD, pulmonary circulation disorders; PEP_ULCER, peptic ulcer disease excluding bleeding; PVD, peripheral vascular disorders; RMDs, rheumatic and musculoskeletal diseases.

diabetes and mental health problems, in keeping with our findings. Our study was not restricted to elderly patients but shows that the mean number of comorbidities is also higher in women at younger ages (25–100 years) while in men, the increased mean number of comorbidities was observed from the age of 50 years.

Patients with RA have also been reported to have a greater prevalence of diabetes mellitus and hypertension, two key risk factors for chronic kidney disease, possibly arising from the use of corticosteroids to manage RA.[30] Some studies show that weight loss is a strong predictor of death in patients with RA[31–33] so it is important to recognise that weight loss is more highly prevalent across all RMDs from our study.

For patients with axial spondyloarthritis identified from health insurance data in Germany, the most prevalent comorbid Elixhauser conditions were 'hypertension' followed by 'depression' and 'chronic pulmonary' disease,[34] illustrating that these comorbid conditions are common regardless of the specific RMD. Based on the findings, it will be important for international guideline committees to assess comorbidities, rather than specific comorbidities.

The European League Against Rheumatism (EULAR) guidelines provide specific recommendations for the management of several comorbidities in individuals with RMDs, including cardiovascular disease, osteoporosis, depression and anxiety.[35] For example, the guidelines recommend that individuals with RMDs and cardiovascular disease should be treated with statins and antiplatelet agents[35]; Individuals with RMDs and osteoporosis should be treated with bisphosphonates and that depression or anxiety should be treated with cognitive

behavioural therapy or antidepressants.[35] However, other comorbidities are not considered specifically. The implications of our findings are that holistic assessment of all comorbidities is required, and our results suggest this is applicable to RMDs as a whole.

### Strengths and limitations

Our study has several strengths: first, to our knowledge, this is the first study to present analysis using different approaches in the large population of Wales in people with and without RMDs using routine primary care data so avoiding biased sampling strategies. Furthermore, the study design minimised bias by matching groups on relevant confounders. Second, our large cohort provides sufficient statistical power to calculate overall and age-specific and sex-specific prevalence rates of individual comorbidities between 2010 and 2019. Finally, in addition to the recognised and frequently reported comorbidities such as cardiovascular risk factors, our findings highlight a number of comorbidities such as deficiency anaemia, fluid and electrolyte disorders and valvular disease that are highly prevalent in people with RMDs but which are under-reported in the literature.

The study replicates and confirms the finding from previous studies, for example, Ziade et al[22] in a 769 Lebanese population, who report the most frequent comorbidities for patients with RMDs were cardiovascular risk factors (hypertension and diabetes), hypercholesterolemia and depression. Consistent with our findings, all reports listed hypertension as the most common individual comorbidity, but we also add renal disease, weight loss, fluid and electrolyte disorders and valvular disease as new pair of comorbidities, that could be tested

**Table 1** ORs (95% CI) for associations of RMDs with specific comorbidities in annual cohorts 2010–2019

| Conditions | 2010 | 2011 | 2012 | 2013 | 2014 | 2015 | 2016 | 2017 | 2018 | 2019 |
|---|---|---|---|---|---|---|---|---|---|---|
| Alcohol abuse | 0.73 (0.58 to 0.93) | 0.75 (0.63 to 0.88) | 0.71 (0.62 to 0.82) | 0.71 (0.63 to 0.80) | 0.68 (0.61 to 0.75) | 0.67 (0.61 to 0.74) | 0.67 (0.61 to 0.73) | 0.66 (0.61 to 0.72) | 0.68 (0.62 to 0.74) | 0.64 (0.59 to 0.69) |
| Arrythmia | 1.06 (0.94 to 1.20) | 1.04 (0.96 to 1.14) | 1.04 (0.97 to 1.12) | 1.09 (1.02 to 1.16) | 1.10 (1.04 to 1.16) | 1.10 (1.05 to 1.16) | 1.15 (1.09 to 1.21) | 1.15 (1.10 to 1.21) | 1.18 (1.13 to 1.24) | 1.13 (1.08 to 1.19) |
| CPD | 1.09 (1.00 to 1.18) | 1.14 (1.08 to 1.21) | 1.13 (1.08 to 1.19) | 1.13 (1.08 to 1.18) | 1.10 (1.06 to 1.15) | 1.08 (1.04 to 1.12) | 1.07 (1.03 to 1.10) | 1.07 (1.04 to 1.11) | 1.03 (1.00 to 1.07) | 1.02 (0.99 to 1.05) |
| CHF | 0.99 (0.85 to 1.16) | 1.00 (0.89 to 1.11) | 1.00 (0.91 to 1.10) | 0.99 (0.91 to 1.07) | 1.03 (0.95 to 1.10) | 0.99 (0.92 to 1.06) | 0.99 (0.93 to 1.06) | 0.99 (0.93 to 1.06) | 0.95 (0.89 to 1.02) | 0.91 (0.85 to 0.97) |
| DEF_ANAEMIA | 1.26 (1.11 to 1.42) | 1.33 (1.22 to 1.45) | 1.36 (1.27 to 1.46) | 1.36 (1.28 to 1.44) | 1.37 (1.29 to 1.44) | 1.39 (1.32 to 1.46) | 1.37 (1.31 to 1.44) | 1.32 (1.26 to 1.39) | 1.32 (1.26 to 1.38) | 1.30 (1.25 to 1.36) |
| Depression | 1.17 (1.07 to 1.27) | 1.13 (1.07 to 1.20) | 1.08 (1.03 to 1.14) | 1.07 (1.03 to 1.12) | 1.07 (1.03 to 1.11) | 1.03 (0.99 to 1.06) | 1.01 (0.98 to 1.05) | 1.00 (0.97 to 1.03) | 0.96 (0.94 to 0.99) | 0.95 (0.92 to 0.98) |
| DIAB | 1.13 (1.02 to 1.26) | 1.17 (1.08 to 1.26) | 1.16 (1.09 to 1.23) | 1.17 (1.11 to 1.23) | 1.13 (1.08 to 1.19) | 1.15 (1.10 to 1.21) | 1.13 (1.09 to 1.18) | 1.13 (1.09 to 1.18) | 1.13 (1.09 to 1.18) | 1.14 (1.10 to 1.18) |
| DIAB_COMP | 0.89 (0.76 to 1.05) | 0.86 (0.77 to 0.96) | 0.87 (0.79 to 0.95) | 0.92 (0.84 to 1.00) | 0.94 (0.87 to 1.01) | 0.92 (0.86 to 0.98) | 0.92 (0.86 to 0.98) | 0.93 (0.87 to 0.99) | 0.88 (0.83 to 0.94) | 0.90 (0.85 to 0.95) |
| Drug abuse | 0.91 (0.66 to 1.25) | 0.68 (0.53 to 0.86) | 0.81 (0.67 to 0.98) | 0.80 (0.68 to 0.94) | 0.75 (0.65 to 0.87) | 0.77 (0.67 to 0.89) | 0.79 (0.69 to 0.90) | 0.79 (0.70 to 0.89) | 0.75 (0.67 to 0.85) | 0.73 (0.65 to 0.82) |
| FED | 1.26 (1.07 to 1.47) | 1.28 (1.14 to 1.43) | 1.26 (1.14 to 1.39) | 1.27 (1.17 to 1.38) | 1.21 (1.12 to 1.31) | 1.21 (1.13 to 1.30) | 1.19 (1.11 to 1.27) | 1.22 (1.14 to 1.30) | 1.18 (1.10 to 1.26) | 1.16 (1.09 to 1.24) |
| Hypertension | 1.02 (0.94 to 1.12) | 1.05 (0.99 to 1.12) | 1.06 (1.01 to 1.12) | 1.05 (1.00 to 1.09) | 1.04 (1.00 to 1.08) | 1.01 (0.97 to 1.04) | 1.02 (0.99 to 1.06) | 1.01 (0.98 to 1.06) | 1.03 (1.00 to 1.06) | 1.04 (1.01 to 1.07) |
| Hypothyroid | 1.34 (1.19 to 1.50) | 1.19 (1.10 to 1.29) | 1.18 (1.10 to 1.26) | 1.21 (1.14 to 1.29) | 1.18 (1.12 to 1.24) | 1.20 (1.14 to 1.26) | 1.24 (1.19 to 1.30) | 1.25 (1.20 to 1.31) | 1.25 (1.20 to 1.31) | 1.27 (1.22 to 1.32) |
| Liver disease | 0.83 (0.62 to 1.12) | 1.03 (0.84 to 1.26) | 0.97 (0.83 to 1.14) | 1.03 (0.90 to 1.18) | 1.10 (0.98 to 1.24) | 1.11 (1.00 to 1.24) | 1.13 (1.02 to 1.25) | 1.09 (0.99 to 1.21) | 1.01 (0.92 to 1.12) | 1.03 (0.94 to 1.14) |
| Obesity | 1.11 (1.00 to 1.23) | 1.08 (1.00 to 1.16) | 1.02 (0.96 to 1.08) | 1.06 (1.01 to 1.12) | 1.08 (1.04 to 1.13) | 1.08 (1.04 to 1.13) | 1.08 (1.04 to 1.13) | 1.08 (1.04 to 1.12) | 1.09 (1.05 to 1.13) | 1.07 (1.03 to 1.11) |
| OND | 0.91 (0.82 to 1.01) | 0.89 (0.83 to 0.95) | 0.94 (0.89 to 1.00) | 0.91 (0.87 to 0.96) | 0.92 (0.88 to 0.96) | 0.90 (0.86 to 0.96) | 0.91 (0.87 to 0.95) | 0.89 (0.86 to 0.93) | 0.87 (0.84 to 0.91) | 0.84 (0.81 to 0.88) |
| PEP_ULCER | 1.02 (0.84 to 1.23) | 1.04 (0.91 to 1.19) | 1.06 (0.95 to 1.19) | 1.04 (0.95 to 1.15) | 1.03 (0.95 to 1.12) | 1.05 (0.97 to 1.14) | 1.07 (0.99 to 1.16) | 1.05 (0.97 to 1.13) | 1.00 (0.93 to 1.07) | 0.98 (0.91 to 1.05) |
| PVD | 0.88 (0.73 to 1.05) | 0.92 (0.82 to 1.05) | 0.93 (0.84 to 1.03) | 0.99 (0.90 to 1.08) | 0.97 (0.89 to 1.06) | 0.98 (0.91 to 1.06) | 0.93 (0.86 to 1.00) | 0.95 (0.88 to 1.02) | 0.89 (0.82 to 0.95) | 0.88 (0.82 to 0.94) |
| PCD | 1.39 (1.12 to 1.73) | 1.33 (1.15 to 1.55) | 1.30 (1.14 to 1.47) | 1.24 (1.11 to 1.47) | 1.25 (1.13 to 1.38) | 1.25 (1.13 to 1.37) | 1.24 (1.14 to 1.36) | 1.25 (1.15 to 1.37) | 1.11 (1.02 to 1.21) | 1.15 (1.06 to 1.25) |
| Renal failure | 1.35 (1.22 to 1.50) | 1.31 (1.22 to 1.41) | 1.28 (1.20 to 1.36) | 1.24 (1.17 to 1.31) | 1.21 (1.15 to 1.31) | 1.22 (1.16 to 1.27) | 1.16 (1.11 to 1.22) | 1.13 (1.08 to 1.18) | 1.14 (1.09 to 1.19) | 1.13 (1.09 to 1.18) |
| Cancer | 0.92 (0.84 to 1.02) | 0.93 (0.87 to 0.99) | 0.90 (0.85 to 0.95) | 0.90 (0.85 to 0.94) | 0.89 (0.85 to 0.93) | 0.89 (0.85 to 0.93) | 0.87 (0.84 to 0.90) | 0.85 (0.82 to 0.89) | 0.84 (0.81 to 0.87) | 0.81 (0.78 to 0.84) |
| Valvular disease | 1.16 (1.00 to 1.35) | 1.27 (1.15 to 1.41) | 1.20 (1.10 to 1.31) | 1.26 (1.17 to 1.36) | 1.24 (1.16 to 1.33) | 1.22 (1.15 to 1.31) | 1.17 (1.10 to 1.25) | 1.17 (1.10 to 1.24) | 1.15 (1.08 to 1.22) | 1.13 (1.07 to 1.20) |
| Weight loss | 1.20 (1.03 to 1.39) | 1.30 (1.17 to 1.45) | 1.26 (1.16 to 1.38) | 1.30 (1.20 to 1.40) | 1.24 (1.16 to 1.33) | 1.22 (1.14 to 1.30) | 1.20 (1.13 to 1.28) | 1.26 (1.19 to 1.34) | 1.20 (1.13 to 1.27) | 1.16 (1.09 to 1.23) |

CHF, congestive heart failure; CPD, chronic pulmonary disease; DEF_ANAEMIA, deficiency anaemia; DIAB, diabetes; DIAB_COMP, diabetes complicated; FED, fluid and electrolyte disorder; OND, other neurological disorders; PCD, pulmonary circulation disorders; PEPTIC_ULCER, peptic ulcer disease excluding bleeding; PVD, peripheral vascular disorders.

directly in other populations with data available. It is essential to recognise that our focus is on investigating patterns of multimorbidity rather than establishing causal relationships.

Our study is a first step towards service transformation for patients with multimorbid RMD. It does not provide directly actionable recommendations but can be used as a starting point to consider creating dedicated pathways for patients with comorbidities that commonly co-occur with RMD, similar to EULAR guidelines.

The aetiology of RMD in Wales is likely comparable to other UK nations with similar access to the NHS and genetic associations related to RMD. Consequently, we anticipate that the risk factors and the pattern of coexisting health conditions will be similar. However, when applying this method to other countries, it is essential to consider that the specific comorbidities may differ based on the unique health characteristics of each country. To enhance the generalisability, the findings should be validated in diverse healthcare systems and patient populations.

The findings could inform policies around resource allocation, healthcare planning and the development of integrated care models for individuals with RMDs and comorbidities; while that is beyond the scope of the current work, our findings will provide valuable information from a large population, using robust and established methods to highlight that patients with RMDs have a higher burden of comorbidities that start to occur at younger ages, particularly in women.

The SAIL Databank holds records from administrative sources within the NHS; efforts to link non-health data such as socioeconomic status, education and lifestyle factors are underway, but it is likely to be some years before such linkage is available. This is a limitation of the current analysis, as well as a focus for future work.

A further limitation is that we have analysed only pairs of diseases, but higher-order combinations of diseases probably interact in more complicated ways. While pairwise comparison is a simple and easy-to-understand method, advanced statistical methods such as network analysis,[36] and clustering algorithms can provide a more nuanced understanding of comorbidities by considering the interactions between multiple diseases and could be applied in future work.

Further research is required to explore potential variation within subpopulations, such as differences among rural and urban areas, and we would like to note that our point about generalisability of these findings with regard to ethnic background are based on the fact that the population of Wales are majorly white (above 80%).[37] In terms of the application of these findings, it should be noted that the prevalence measure was based on Elixhauser conditions; there is no established gold standard list of diagnoses used to define multimorbidity and definitions of multimorbidity vary in the medical literature,[38] making direct comparisons across studies and different populations more challenging. However, separately both the Charlson and Elixhauser indices are often used[39] and

the Cambridge multimorbidity score has been published recently[38] and there is some overlap between these indices in defining comorbidity.

## CLINICAL IMPLICATIONS

We emphasise early detection of RMDs and associated conditions, especially in younger women. Healthcare practitioners should facilitate patient access to community resources such as exercise programmes and support groups to enhance overall well-being and address depression in the age range of 16 to 50 years based on our study findings. We also introduce focused screening initiatives for prevalent comorbidities linked to RMDs, including renal disease, weight loss and fluid and electrolyte disorder and valvular disease.

Considering the high prevalence of RMDs in the population, the discovery that patients face a significantly elevated risk of various co-morbidities is crucial. This finding could guide targeted screening efforts for primary and secondary prevention in clinical settings. This is particularly relevant for the common comorbidities such as hypertension, where multiple treatments are available and screening at healthcare visits in this high-risk population would provide an opportunity to introduce prevention measures for subsequent conditions such as cardiovascular disease and stroke. The finding that risks are increased from a younger age in patients with RMD and in women implies that screening should be extended to the whole RMD population rather than the general population screening, which focusses on older age groups.

## CONCLUSION

In conclusion, we provide the first detailed 10-year epidemiological analysis of multimorbidity burden in people with RMDs in comparison with individuals without RMDs using Welsh primary care data. We found a substantially higher prevalence and burden of multimorbidity in patients with RMDs relative to those without, suggesting that RMDs patients represent a high-risk group for multimorbidity. A greater understanding of multimorbidity could aid clinicians in developing and delivering comprehensive treatment pathways and to action primary and secondary prevention of comorbidities in people with RMDs.

This higher mean number of comorbidities is seen from the age of 25 in women and 50 in men. Further research is required to determine whether targeted interventions could reduce overall disease burden from multimorbidity in that group. This study depends on diagnoses reported in electronic health records, it is possible that patients with undiagnosed conditions, such as those who did not visit a general practitioner, were missed in our study.

**Author affiliations**
[1]Centre for Genetics and Genomics Versus Arthritis, Centre for Musculoskeletal Research, Faculty of Biology, Medicine and Health, Manchester Academic Health Science Centre, The University of Manchester, Manchester, UK

[2]Population Data Science, Swansea University Medical School, Swansea, UK
[3]Division of Informatics, Imaging and Data Science, School of Health Science, Faculty of Biology, Medicine and Health, University of Manchester, Manchester, UK
[4]NIHR Manchester Musculoskeletal Biomedical Research Unit, Central Manchester NHS Foundation Trust, Manchester Academic Health Science Centre, Manchester, UK
[5]The Healthcare Improvement Studies Institute (THIS Institute), Department of Public Health and Primary Care, University of Cambridge, Cambridge, UK

**Acknowledgements** The views expressed are those of the authors and not necessarily of the NHS, the National Institute for Health Research or the Department of Health. The authors would also like to acknowledge all data providers who make anonymised data available for research. All research conducted has been completed under the permission and approval of the SAIL independent Information Governance Review Panel (IGRP) project number 0911. This work uses data provided by patients and collected by the NHS as part of their care and support. The authors would like to acknowledge all data providers who make anonymised data available for research. We also acknowledge help and advice from Dr Salwa S Zghebi from School of Health Science, Manchester University, UK.

**Contributors** FJ-n: Methodology, software, formal analysis and writing original draft. RB: Formal analysis and writing—review and editing. JL: Data curation and writing—review and editing. AA, TBD, NA, JR, AW and GPM: Writing—review and editing. JB and AB: Supervision and writing—review and editing. RL: Project administration and writing—review and editing. NP: Project administration, supervision and writing—review and editing.FJ-n had full access to all study data, performed all the statistical analyses, supervised by NP and AB and takes responsibility for the integrity of the data and the accuracy of data analyses. All authors contributed to interpretation of data and revised the paper for important intellectual content and agreed on the final version before submission.

**Funding** The datasets used in this study were supported by grants cofunded by Medical Research Council (MRC) and National Institute for Health Research (NIHR) (grant number: MR/S027750/1); and supported by Health Data Research UK (grant number: HDR-9006), which receives its funding from the UK Medical Research Council, Engineering and Physical Sciences Research Council, Economic and Social Research Council, Department of Health and Social Care (England), Chief Scientist Office of the Scottish Government Health and Social Care Directorates, Health and Social Care Research and Development Division (Welsh Government), Public Health Agency (Northern Ireland), British Heart Foundation and the Wellcome Trust. This work was supported by the ADR Wales programme of work. The ADR Wales programme of work is aligned to the priority themes as identified in the Welsh Government's national strategy: Prosperity for All. ADR Wales brings together data science experts at Swansea University Medical School, staff from the Wales Institute of Social and Economic Research, Data and Methods (WISERD) at Cardiff University and specialist teams within the Welsh Government to develop new evidence which supports Prosperity for All by using the SAIL Databank at Swansea University, to link and analyse anonymised data. ADR Wales is part of the Economic and Social Research Council (part of UK Research and Innovation) funded ADR UK (grant ES/S007393/1). This research was partially funded by the NIHR's Manchester Biomedical Research Centre. The views expressed are those of the author(s) and not necessarily those of the NHS, the National Institute for Health research or the Department of Health and Social Care. The work is supported by the Centre for Genetics and Genomics Versus Arthritis (UK grant number 21754). AB is an NIHR Senior Investigator. FJ-n's research is supported by an MRC/University of Manchester Skills Development Fellowship (grant number MR/R016615).

**Competing interests** None declared.

**Patient and public involvement** Patients and/or the public were not involved in the design, or conduct, or reporting, or dissemination plans of this research.

**Patient consent for publication** Not applicable.

**Ethics approval** This study involves human participants. Approval for the use of anonymised data in this study, provisioned within the Secure Anonymised Information Linkage (SAIL) Databank, was granted by an independent Information Governance Review Panel (IGRP) under project 0911. The IGRP has a membership comprised of senior representatives from the British Medical Association, the National Research Ethics Service (NRES), Public Health Wales and Digital Health and Care Wales and members of the public. The SAIL Databank is compliant with General Data Protection Regulations and the UK Data Protection Act. SAIL provides access to deidentified data and, as such, informed consent was not applicable, as per NRES guidance. All research was performed in accordance with SAIL Databank processes and guidelines and relevant regulations. Participants gave informed consent to participate in the study before taking part.

**Provenance and peer review** Not commissioned; externally peer reviewed.

**Data availability statement** Data are available on reasonable request. The data used in this study are available in the SAIL Databank at Swansea University, Swansea, UK, but as restrictions apply they are not publicly available. All proposals to use SAIL data are subject to review by an independent Information Governance Review Panel (IGRP). Before any data can be accessed, approval must be given by the IGRP. The IGRP gives careful consideration to each project to ensure proper and appropriate use of SAIL data. When access has been granted, it is gained through a privacy protecting safe haven and remote access system referred to as the SAIL Gateway. SAIL has established an application process to be followed by anyone who would like to access data via SAIL at https://www.saildatabank.com/application-process.

**ORCID iDs**
Farideh Jalali-najafabadi http://orcid.org/0000-0003-4895-4578
Jane Lyons http://orcid.org/0000-0002-4407-770X
Ashley Akbari http://orcid.org/0000-0003-0814-0801
Alan Watkins http://orcid.org/0000-0003-3804-1943
Glen Philip Martin http://orcid.org/0000-0002-3410-9472
Ronan A. Lyons http://orcid.org/0000-0001-5225-000X
Anne Barton http://orcid.org/0000-0003-3316-2527

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
