## [Reviewer comments · BMJ Open]

ARTICLE DETAILS

TITLE (PROVISIONAL)	10-year multi-morbidity patterns among people with and without rheumatic and musculoskeletal diseases: An observational cohort study using linked electronic health records from Wales, UK
AUTHORS	Jalali-najafabadi, Farideh; Bailey, Rowena; Lyons, Jane; Akbari, Ashley; Badhafari, Tthamer; Azadbakht, Narges; Rafferty, James; Watkins, Alan; Martin, Glen; Bowes, John; Lyons, Ronan; Barton, Anne; Peek, N

VERSION 1 – REVIEW

REVIEWER	Fatemi Aghda1, Seyed Ali Iran University of Medical Sciences
REVIEW RETURNED	28-Sep-2023

GENERAL COMMENTS	Thank you for your efforts in conducting this study. This paper has the potential to be accepted, but some important points have to be clarified or fixed before we can proceed, and positive action can be taken. 1- Move beyond pairs of diseases and employ advanced statistical techniques, such as network analysis or clustering algorithms, to explore complex relationships among co-morbidities. This will provide a more nuanced understanding of how multiple conditions interact. 2- To enhance generalizability, validate the findings by conducting similar studies in different regions or countries with diverse healthcare systems and patient populations. This can help identify common trends and regional variations. 3-The paper primarily presents associations between RMDs and co-morbidities but does not establish causality. It's unclear whether RMDs directly contribute to the development of co-morbidities or if they share common risk factors. Incorporate longitudinal data and causal inference methods to investigate whether RMDs play a causal role in the development of co-morbidities or if they share common risk factors. Establishing causality can inform targeted interventions.
---

	4- The study does not provide insights into potential interventions or strategies for managing co-morbidities in individuals with RMDs. While it identifies the prevalence of co-morbidities, it falls short of offering actionable recommendations for healthcare practitioners. 5- Discuss practical clinical implications and recommendations based on the study's findings. Provide guidance for healthcare practitioners on managing co-morbidities in individuals with RMDs and highlight opportunities for early intervention and prevention. 6- The paper treats RMDs as a single group without distinguishing between different types or severity levels of RMDs. This oversimplification may mask variations in co-morbidity prevalence across specific RMD subtypes. 7- Although the paper discusses the generalizability of its findings to the Welsh population, it does not explore potential variations within subpopulations, such as differences among rural and urban areas or ethnic groups. Explore variations in co-morbidity patterns within subpopulations, such as differences between rural and urban areas, socioeconomic groups, or ethnicities. Tailor recommendations to address specific needs. 8- The research does not account for social determinants of health, such as socioeconomic status, education, and lifestyle factors, which can significantly influence co-morbidity prevalence. 9- While the paper notes that co-morbidities are more prevalent in women and that the mean number of co-morbidities varies with age, it does not delve into the underlying reasons or explore potential gender- or age-specific risk factors. 10- Discuss the policy implications of the findings, particularly in terms of resource allocation, healthcare planning, and the development of integrated care models for individuals with RMDs and co-morbidities. 11-Considering that we are at the end of 2023, it is better to update the references. For example, reference number 7 is 1986!!!!!! (Isn't another study done)
--	---

REVIEWER	Khanh Le, Nguyen Quoc Taipei Medical University
REVIEW RETURNED	03-Oct-2023

GENERAL COMMENTS	1. The study uses Elixhauser coding algorithms downloaded from a supplementary source. It is crucial to provide information on the validation or reliability of these coding algorithms, as the accuracy of comorbidity identification heavily relies on the robustness of these coding systems. 2. While the study provides an overview of the Welsh Multi-morbidity e-Cohort (WMC), it lacks details about the representativeness of the
--

	cohort in relation to the broader population. 3. The study defines the follow-up period until December 31, 2019. However, it does not elaborate on why this specific duration was chosen and whether it is sufficient to capture long-term outcomes related to rheumatoid arthritis/collagen vascular diseases (RMDs). 4. The criteria for matching patients in the RMDs and non-RMDs cohorts are mentioned, but the rationale for the ± 2 years age difference and the selection of the diagnostic period (2010-2019) could be explained further. Additionally, details on how RMDs diagnoses were made and recorded would enhance the study's transparency. 5. While the study defines overall morbidity and multi-morbidity, the exact definitions of these terms and the methods used to measure them are not explicitly stated. 6. The authors used four approaches in statistical analysis but lacks a clear and concise explanation of each approach. 7. Some methodological details, such as how missing data were handled or any sensitivity analyses conducted, are not explicitly mentioned. 8. While the study provides a comprehensive methodology, there is limited discussion of potential limitations. 9. The study mentions that results for seven co-morbidities with fewer than 5 cases are not reported due to statistical disclosure control. Provide a brief explanation or reference about what statistical disclosure control entails to ensure transparency and help readers understand the limitations imposed. 10. The authors discussed odds ratios (ORs) lacks information on the statistical methods used for these calculations. They should provide a brief overview of the statistical model or regression analysis employed to derive these odds ratios. 11. The study excludes reporting on co-morbidities with fewer than 5 cases. They should provide a brief justification for this exclusion criterion. Additionally, discuss any potential implications or limitations associated with this decision to ensure transparency in reporting. 12. While odds ratios are presented, there is limited interpretation or discussion of their clinical significance. 13. The study mentions that the mean number of comorbidities increased with age, but more details on the age-specific analysis, especially in relation to specific co-morbidities, would provide a more comprehensive understanding of the impact of age on the results. 14. While the study mentions that odds ratios were generally stable over the study period, consider providing a brief discussion on any observed temporal trends or variations, especially if there are notable changes in specific co-morbidities over the years.
--	--

VERSION 1 – AUTHOR RESPONSE

*** Reviewer #1 (statistical review)

R1.1. Move beyond pairs of diseases and employ advanced statistical techniques, such as network analysis or clustering algorithms, to explore complex relationships among comorbidities. This will provide a more nuanced understanding of how multiple conditions interact.

Thank you for your suggestion for using advanced statistical methods. Our objective in this paper is to compare pattern of multi-morbidity between people with and without rheumatic and

musculoskeletal diseases (RMDs) use an established statistical method for the pair of the diseases. [AB3][NP4]

The more advanced statistical methods would go beyond the research objective. That is not to say that they are not useful, but simply out of scope for our paper. We have added the below statement as the limitation of our study in Discussion part.

“While pairwise comparison is a simple and easy-to-understand method, advanced statistical methods such as network analysis and clustering algorithms can provide a more nuanced understanding of comorbidities by considering the interactions between multiple diseases “.

R1.2. To enhance generalizability, validate the findings by conducting similar studies in different regions or countries with diverse healthcare systems and patient populations. This can help identify common trends and regional variations.

We agree with the reviewer and would welcome the opportunity to collaborate with researchers in other regions and countries. This work was carried out using the SAIL Databank which is world-leading resource with research ready data and computational infrastructure. [AB5][NP6]

This study replicates the findings from Ziyade et, that report the most frequent comorbidities for RMDs patients as cardiovascular risk factor (hypertension, and diabetes) in Lebanese population as was highlighted in the discussion before. Consistent with our findings, all reports listed hypertension as the most common individual comorbidity, but we also add renal disease, weight loss, fluid and electrolyte disorders and valvular as the new pair of co-morbidities.

We have added the below sentences as the as a limitation to the current study in the Discussion. “To enhance the generalizability the finding should be validated in diverse healthcare system and patient population”.

R1.3. The paper primarily presents associations between RMDs and co-morbidities but does not establish causality. It's unclear whether RMDs directly contribute to the development of co-morbidities or if they share common risk factors. Incorporate longitudinal data and causal inference methods to investigate whether RMDs play a causal role in the development of co-morbidities or if they share common risk factors.

Establishing causality can inform targeted interventions.

Thanks for this interesting point. This retrospective analysis of observational cohort data is not suited for causal associations; however, as the reviewer implies, this would be an interesting next step in future work. In the Discussion, we make recommendations for prospective data collection that would facilitate the evaluation of causality [AB7].

R1.4. The study does not provide insights into potential interventions or strategies for managing co-morbidities in individuals with RMDs. While it identifies the prevalence of co-morbidities, it falls short of offering actionable recommendations for healthcare practitioners.

There exist EULAR guidelines about interpretation of cardiovascular risk scores in patients with RA, for example: Cardiovascular risk prediction models should be adapted for patients with RA (Rheumatoid arthritis) by a 1.5 multiplication factor, this is not already included in the risk algorithm.

[AB8] We have added the below statement in discussion section.

“The European League Against Rheumatism (EULAR) guidelines provide specific recommendations for the management of several co-morbidities in individuals with RMDs, including cardiovascular disease, osteoporosis, depression, and anxiety(Raechel A.DAmarrell). However, other co-morbidities are not considered specifically. The implications of the findings is that holistic assessment of all co-morbidities is required and across RMDs as a whole.

For example, the guidelines recommend that individuals with RMDs and cardiovascular disease should be treated with statins and antiplatelet agents¹. The guidelines also recommend that individuals with RMDs and osteoporosis should be treated with bisphosphonates¹. The guidelines suggest that individuals with RMDs and depression or anxiety should be treated with cognitive behavioural therapy or antidepressants¹.

It is important to note that the guidelines are intended to be used as a tool to aid clinical decision-making and are not intended to replace clinical judgment¹. Therefore, it is important to consult with a healthcare professional to determine the most appropriate management strategy for an individual with RMDs and co-morbidities. ”

R1.5. Discuss practical clinical implications and recommendations based on the study's findings. Provide guidance for healthcare practitioners on managing co-morbidities in individuals with RMDs and highlight opportunities for early intervention and [FJ9][m10] prevention.

As above, guidelines exist on the management of cardiovascular risk in patients with RA, PsA and axSpA and we have now included this in the manuscript. However, based on the findings, it will be important for international guideline committees to assess co-morbidities as a whole, rather than specific co-morbidities and this has now also been highlighted in the Discussion.

R1.6. The paper treats RMDs as a single group without distinguishing between different types or severity levels of RMDs. This oversimplification may mask variations in comorbidity prevalence across specific RMD subtypes.

As you have suggested, we have done the analysis for the RMDs subtype. The subtypes are as follow.

RA (Rheumatoid arthritis), CTD - connective tissue diseases, (PMR) polymyalgia rheumatica and giant cell arteritis cases and JIA: juvenile arthritis. For all these subtypes likewise RMDs, we have observed the most common co-morbidities as cardiovascular risk factors, hypertension and diabetes without complications.

The annual prevalence of Elixhauser co-morbidities for each RMDs subtype is presented in Figure 1 supplementary.

R1.7. Although the paper discusses the generalizability of its findings to the Welsh population, it does not explore potential variations within subpopulations, such as differences among rural and urban areas or ethnic groups. Explore variations in comorbidity patterns within subpopulations, such as differences between rural and urban areas, socioeconomic groups, or ethnicities. Tailor recommendations to address specific needs.

Thank you for this interesting suggestion; whilst we did not set out to explore variations within subpopulations, this could be a focus of future work and we have included the below statement in the Discussion. We have not explored the variations in comorbidity patterns within subpopulations, such as differences between rural and urban areas, socioeconomic groups, or ethnicities and we consider this as the future work. According to a report by the Office for National Statistics, there are significant economic inequalities in the UK, with London and the Southeast having the highest income and productivity in 2018. The same report also provides and interactive map of economic inequality in the

UK. A report by the UK government highlights the need to understand the interplay of social, economic, biological, and pre-pandemic health risks that vary across ethnic groups when exploring the causes of ethnic inequalities in COVID-19.() Another report by the same organization discusses ethnic disparities in the major causes of mortality and their risk factors.()
A study published from (Sheri etl) found that comorbidities were more common in the rural patient population, as were the number of comorbidities per patient.

R1.8. The research does not account for social determinants of health, such as socioeconomic status, education, and lifestyle factors, which can significantly influence co-morbidity prevalence.

This data used the SAIL databank which holds records from administrative sources within the NHS; efforts to link non-health data such as education and lifestyle factors is underway but it is likely to be some years before such linkage is available. The point is an important one and we have included further Discussion to highlight this as a limitation of the current analysis, as well as a focus for future work.

R1.9. While the paper notes that co-morbidities are more prevalent in women and that the mean number of co-morbidities varies with age, it does not delve into the underlying reasons or explore potential gender- or age-specific risk factors.

As you have suggested, we have done the age-specific analysis, especially in relation to the specific co-morbidities for RMDs and comparators. The results are presented in Figure 2 supplementary pages (2-6). Depression was observed as the most prevalent condition in age group [16-25] and [25-50] for RMDs and non-RMDs. Diabetes without complications for RMDs was the second prevalent condition. Other neurological disorders (OND) and cancer were the second and third prevalent condition for non RMDs respectively.

R1.10. Discuss the policy implications of the findings, particularly in terms of resource allocation, healthcare planning, and the development of integrated care models for individuals with RMDs and co-morbidities.

The findings could inform policies around resource allocation, healthcare planning, and the development of integrated care models for individuals with RMDs and co-morbidities; whilst that is beyond the scope of the current work, our findings will provide valuable information from a large population, using robust and established methods to highlight that patients with RMDs have a higher burden of co-morbidities that start to occur at younger ages, particularly in women.[NP11]

R1.11. Considering that we are at the end of 2023, it is better to update the references. For example, reference number 7 is 1986!!!!!!! (Isn't another study done)

1. Agreed and substituted [NP12][m13] with the below reference.

Palazzo, C., Ravaud, J.-F., Papelard, A., Ravaud, P. & Poiraudou, S.
The burden of musculoskeletal conditions. *PLoS one* 9, e90633 (2014)

*** Reviewer #2:

Abstract:

R2.1. The study uses Elixhauser coding algorithms downloaded from a supplementary source. It is crucial to provide information on the validation or reliability of these coding algorithms, as the accuracy of comorbidity identification heavily relies on the robustness of these coding systems.

Thank you. We agree that validity is an important issue. International Classification of Diseases 10th revision (ICD-10) codes, OPCS Classification of Interventions and Procedures version 4 (OPCS-4) codes and primary care Read version 2 (CTV2) codes were used to identify Elixhauser morbidities in our dataset. The Elixhauser coding algorithms used to establish the presence of each Elixhauser morbidity were downloaded from the supplementary material of Metcalfe et al (reference 26) which has evidenced content and construct [AB14] validity. The full list of morbidities are listed in Table 1 supplementary and resulted in a set of 28 comorbidities and Rheumatoid arthritis/collagen vascular diseases (RMDs).

Reference 26 – Metcalfe et al developed the Elixhauser coding algorithms for READ codes used in UK General Practice. They had three aims:

“The aims of this study were to: (1) develop coding algorithms for calculating CCI and EM in Read-coded databases, (2) describe the comorbidity characteristics of a hip fracture cohort with matched controls, and (3) compare the predictive properties of the CCI (both original and modified versions) and the EM”.

They took a considered approach to content and criterion (predictive) validity.

“Two clinicians independently used the exploded ICD-9-CM text codes to search all 111,929 Read terms within the CPRD Medical Dictionary with discussion and resolution of any differences”.

“In addition, the online ClinicalCodes Repository [19] was manually searched for all pre-existing Read code lists that pertained to each comorbidity category. Lists from 12 studies [20,21,22,23,24,25,26,27,28,29,30] were included from the ClinicalCodes Repository in addition to the CCI list previously developed by Khan et al [13]. The outcome of this process was that between two and six independent Read code lists were generated for each comorbidity category. The two clinicians then resolved discrepancies through discussion and with advice from sub-specialists where appropriate. A single list was generated for each comorbidity measure and duplicate entries deleted. A final logic check was performed by a single clinician”.

This approach covers the content validity of the conditions and codes included through the exhaustive search of READ terms. They also demonstrate criterion validity by showing better predictive properties (higher AUROC) of the EM compared with CCI in both diseased (i.e. hip fracture) and non-diseased (age- and sex-matched control) populations for 30 and 365 day mortality outcomes

R2.2. While the study provides an overview of the Welsh Multi-morbidity e-Cohort (WMC), it lacks details about the representativeness of the cohort in relation to the broader population. [NP15][m16]

The WMC is based on the 83% of the Welsh population covered by GP practices in Wales that contribute to the SAIL databank. A report by Cary Jones in 2022-03-30 (I am not sure, how I should refer to that as it is an internal document and has not published!) showing how this compares to the 100% of the population – the distributions of sex and age-group are identical and the distribution of deprivation scores are very similar.

R2.3. The study defines the follow-up period until December 31, 2019. However, it does not elaborate on why this specific duration was chosen and whether it is sufficient to capture long-term outcomes related to rheumatoid arthritis/collagen vascular diseases (RMDs). [NP17]

[AB18]The study end date of 31st December 2019 was chosen as at the time of cohort creation this was the coverage of data that was available and allowed for a 20-year follow up. The study end date was based on data availability at time of cohort creation and has allowed for a long follow up of the population of Wales. The creation of cohort has been explained in Jane et al reference number 27.

R2.4. The criteria for matching patients in the RMDs and non-RMDs cohorts are mentioned, but the rationale for the ± 2 years age difference and the selection of the diagnostic period (2010-2019) could be explained further. Additionally, details on how RMDs diagnoses were made and recorded would enhance the study's transparency[NP19]

Thank you we have added a sentence to section 4.2.2 "+- 2 year time period was used for matching to increase the comparator sample after discussion with clinicians that age groups within this interval are clinically comparable[FJ20]". Read codes version 2 for GP records were used to identify patient records that had a RMD diagnosis recorded.

R2.5. While the study defines overall morbidity and multi-morbidity, the exact definitions of these terms and the methods used to measure them are not explicitly stated.

Thank you, in the introduction we define multi-morbidity as existence of multiple long-term conditions (MLTCs) in a single individual. Morbidity refers to individual long-term morbid conditions. We have now clarified this in the manuscript.

R2.6. The authors used four approaches in statistical analysis but lacks a clear and concise explanation[FJ21] of each approach[NP22]

The description of the four statistical approaches has been edited to clarify the four approaches in relation to the specific study objectives as follows.

Descriptive statistics of the differences between the cohorts are reported as the total percentage of multi-morbid persons (two or conditions), the percentage distribution of the total number of comorbidities and the mean number of co-morbidities by sex and age-group. To explore the patterns of multimorbidity, the data were analysed using four approaches:

Temporal trends for specific Elixhauser conditions were explored using prevalence rates. Annual prevalence rates were derived for each cohort over the 10-year period and were calculated separately for each comorbidity by dividing the number of persons diagnosed in or before the study year by the total number of persons contributing throughout the same study year. Corresponding 95% confidence intervals were calculated.

The prevalence rates were also calculated stratified by (1) 20-year age groups, (2) sex with RMDs and matched comparator patients in the last study year.

The patterns of morbidity conditions were examined using pairwise associations between all co-morbidities. The percentage of patients with both co-morbidities was calculated for each pair of conditions for each cohort.

logistic regression models were used to explore association between the presence/absence of each co-morbidity and a diagnosis of RMD. Conditional models, adjusted for age and sex, were fitted to estimate differences between cohorts with associations with Elixhauser co-morbidities. The coefficients were exponentiated and reported as odds ratios (ORs) along with corresponding 95% confidence intervals (CIs). Regression models restricted to males and females were also fitted to calculate sex-specific estimates.

R2. 7. Some methodological details, such as how missing data were handled or any sensitivity analyses conducted, are not explicitly mentioned.

We have created section 4.2 and the below statement has been added to cover the missing data.

This analysis uses age, sex and the presence or absence of co-morbidities. Age and sex are complete as they are derived from the algorithm creating the anonymised linkage fields (ALFs) from NHS registration data. The presence of co-morbidities is defined by the presence of the relevant codes in the electronic health records. Where a record does not record a condition, e.g.RMDs, it is

assumed that the record is correct and the person does not have RMDs. Hence, there are no missing data.

R2.8. While the study provides a comprehensive methodology, there is limited discussion [FJ23][AB24][NP25] of potential limitations.

We have highlighted the potential limitations in the discussion as follows.

While pairwise comparison is a simple and easy-to-understand method, advanced statistical methods such as network analysis and clustering algorithms can provide a more nuanced understanding of comorbidities by considering the interactions between multiple diseases. To enhance the generalizability the finding should be validated in diverse healthcare system and patient population. Given that retrospective analysis of observational cohort data isn't ideal for determining causal relationships, we're using this set of prospective data to assess causality.

R2.9. The study mentions that results for seven co-morbidities with fewer than 5 cases are not reported due to statistical disclosure control. Provide a brief explanation or reference about what statistical disclosure control entails to ensure transparency and help readers understand the limitations imposed.

Statistical disclosure control rules meant that cells with less than 5 subjects cannot be reported in order to prevent the inadvertent disclosure of individuals. The guidance was developed by the UK's Office of National Statistics as part of its work on protecting privacy through design and statistical disclosure control as demonstrated in their handbook https://ukdataservice.ac.uk/app/uploads/thf_datareport_aw_web.pdf . SAIL follows the ONS disclosure rules. This has now been added to the manuscript [AB26][NP27].

R2.10. The authors discussed odds ratios (ORs) lacks information on the statistical methods used for these calculations. They should provide a brief overview of the statistical model or regression analysis employed to derive these odds ratios.

We have elaborated the methods with this sentence . "The coefficients were exponentiated and reported as odds ratios (ORs) and their corresponding 95% confidence intervals (CIs)" [NP28] And have further revised the structure of the methods section to make it clearer how each statistical analysis relates to the study aims.

R2.11. The study excludes reporting on co-morbidities with fewer than 5 cases. They should provide a brief justification for this exclusion criterion. Additionally, discuss any potential implications or limitations associated with this decision to ensure transparency in reporting.

These patients were part of the study cohort and the analyses, but small cells were suppressed in the presentation of result to protect the privacy of participants. This is not an exclusion criterion. Excluding reporting of co-morbidities with fewer than 5 cases does not affect the presentation of results on annual prevalence of the numbers of co-morbidities as these easily exceed that number. The only effect would be on the age-specific prevalence of specific conditions that are less frequent in younger ages. [NP29] This rule is imposed on anyone who uses the SAIL databank, it is not specific to our study. It is a very common way to protect the privacy of study participants in cohorts of routinely collected healthcare data.

R2.12. While odds ratios are presented, there is limited interpretation or discussion of their clinical significance [FJ30][RB31].

[NP32]

R2.13. The study mentions that the mean number of comorbidities increased with age, but more details on the age-specific analysis, especially in relation to specific co-morbidities, would provide a more comprehensive understanding of the impact of age on the results. [AB33]
We have done the age-specific analysis, especially in relation to the specific co-morbidities for RMDs and comparators. The results are presented in Figure 2 supplementary pages (2-6). Depression was observed as the most prevalent condition in age group [16-25] and [25-50] for RMDs and non-RMDs. Diabetes without complications for RMDs was the second prevalent condition. Other neurological disorders (OND) and cancer were the second and third prevalent condition for non RMDs respectively. This also has been highlighted in section 5.3 page 8.

R2.14. While the study mentions that odds ratios were generally stable over the study period, consider providing a brief discussion on any observed temporal trends or variations, especially if there are notable changes in specific co-morbidities over the years.

[AB34]ORs were generally stable over the study period between 2010 and 2019, with the exception of depression which was found to be significantly higher among patients with RMDs between 2010 and 2015 but was significantly (albeit slightly) lower in 2018 and 2019. This has been highlighted in section 5.5 page 9.

1

[NP1]Still vague about design. I'd suggest to say: "...: an observational cohort study using linked electronic health records from Wales, UK".

It's bit odd to say "linked primary care records" (linked to what?).

[NP2]They are referring to the fact that the STROBE checklist has a number of blank items.

[AB3]I don't think this addresses the reviewer's point. You need to do something or argue that pairwise is better somehow.

[NP4]I agree with Anne. You should argue that the methods that we have used address the research objective, while more advanced statistical methods would go beyond the research objective. That is not to say that they are not useful, but simply out of scope for our paper.

If you want to refer to Jim's work (which I believe you are intending to do here?) you have to be very explicit that that is a separate paper, and cit it. But frankly, I'm afraid that it only muddies the water. The reviewer is focused on your paper.

[AB5]Again, haven't addressed the comment. Could you say that this study replicates previous findings from different countries but also adds new pairs of co-morbidities, which could be tested by others. We have added this as a limitation to the current study in the Discussion.

[NP6]Agree with Anne.

[AB7]Could you say anything about Mendelian randomisation to infer causality - but genetic data not available for SAIL participants? And add to Discussion?

[AB8]Add something to Discussion

[FJ9]Recommend using MDTs to treat combinations of disease rather than silo'd health settings

[m10]Anne, can you please advise, how this should be added to the manuscript? Thanks

[NP11]We should tell the reviewer what we have changed in the manuscript in response to their comment.

[NP12]Tell the reviewer here by which paper you have substituted the original reference, so they don't have to look it up in the manuscript.

[m13]Done!

[AB14] Could you amend here to say content, construct and criterion validity? Could you add that other researchers have used these criteria with refs?

[NP15] I'd suggest to ask Jane Lyons to formulate a response to this comment. You will need to cite Jane's paper.

[m16] Done!

[NP17] Again discuss with Jane and cite her paper. Important to emphasise that we used an existing cohort (WMC) – it was not created for our study.

[AB18] Doesn't answer the question. Need to say something to address eg in Discussion, you could say

Whilst the SAIL databank contains data covering 20 years, this still may not capture long-term outcomes related to RMDs; however, even within the 20 year time-frame, the additional burden of co-morbidities is apparent.

[NP19] Don't forget to address this comment as well.

[FJ20] I should add a sentences

[FJ21] Consider elaborating the purpose of each stats method

[NP22] The reviewer is right that details are lacking. Let's include them in an appendix.

[FJ23] Ask all co-authors to consider limitations of method..

.. maybe add pairwise comparisons?

[AB24] I think the answers to the previous reviewer highlight some of the limitations and address this point?

[NP25] Exactly!

[AB26] methods

[NP27] Always be precise about what you have changed/added, and were exactly in the manuscript. It avoids that reviewers have to go back and forth between response letter and manuscript – or worse, that they have to search through the manuscript.

[NP28] I think that this fits with the general lack of detail on the methods. It can be solved by providing an appendix with a detailed description.

[NP29] I think we should emphasise that this is not an exclusion criterion. These patients were part of the study cohort and the analyses, but small cells were suppressed in the presentation of result to protect the privacy of participants.

We should also emphasise that this rule is imposed on anyone who uses the SAIL databank, it is not specific to our study. It is a very common way to protect the privacy of study participants in cohorts of routinely collected healthcare data.

[FJ30] Something about effect sizes - translating outcomes into rates or absolute differences in prevalence

[RB31] I think you need to add some statistics to the discussion, pick out some of the key results and discuss the magnitude of the effect

[NP32] We should tell the reviewer what we have changed in the manuscript in response to their comment.

[AB33] Can you add as supplementary data?

[AB34] Need to do or say something to address this point.

VERSION 2 – REVIEW

REVIEWER	Khanh Le, Nguyen Quoc Taipei Medical University
REVIEW RETURNED	12-Apr-2024

GENERAL COMMENTS	My previous comments have been addressed.
---